# Adapting an Atmospheric Dispersion Model to Assess the Risk of Windborne Transmission of Porcine Reproductive and Respiratory Syndrome Virus between Swine Farms

**DOI:** 10.3390/v14081658

**Published:** 2022-07-28

**Authors:** Kaushi S. T. Kanankege, Kerryne Graham, Cesar A. Corzo, Kimberly VanderWaal, Andres M. Perez, Peter A. Durr

**Affiliations:** 1Department of Veterinary Population Medicine, College of Veterinary Medicine, University of Minnesota, St Paul, MN 55108, USA; corzo@umn.edu (C.A.C.); kvw@umn.edu (K.V.); aperez@umn.edu (A.M.P.); 2Commonwealth Scientific and Industrial Research Organisation (CSIRO), Australian Centre for Disease Preparedness, Geelong, VIC 3219, Australia; kerryne.graham@csiro.au (K.G.); peter.durr@csiro.au (P.A.D.)

**Keywords:** pig diseases, spatial epidemiology, Lagrangian models, aerial dispersion, TAPPAS, HYSPLIT, airborne, infectious disease modeling

## Abstract

Modeling the windborne transmission of aerosolized pathogens is challenging. We adapted an atmospheric dispersion model (ADM) to simulate the windborne dispersion of porcine reproductive and respiratory syndrome virus (PRRSv) between swine farms. This work focuses on determining ADM applicable parameter values for PRRSv through a literature and expert opinion-based approach. The parameters included epidemiological features of PRRSv, characteristics of the aerosolized particles, and survival of aerosolized virus in relation to key meteorological features. A case study was undertaken to perform a sensitivity analysis on key parameters. Farms experiencing ongoing PRRSv outbreaks were assigned as particle emitting sources. The wind data from the North American Mesoscale Forecast System was used to simulate dispersion. The risk was estimated semi-quantitatively based on the median daily deposition of particles and the distance to the closest emitting farm. Among the parameters tested, the ADM was most sensitive to the number of particles emitted, followed by the model runtime, and the release height was the least sensitive. Farms within 25 km from an emitting farm were at the highest risk; with 53.66% being within 10 km. An ADM-based risk estimation of windborne transmission of PRRSv may inform optimum time intervals for air sampling, plan preventive measures, and aid in ruling out the windborne dispersion in outbreak investigations.

## 1. Background

With an estimated annual cost of $664 million to the U.S. swine industry, Porcine reproductive and respiratory syndrome virus (PRRSv) is the costliest endemic swine pathogen today [1,2] The costs are related to impairment in the breeding and growth of pigs, the cost of biosecurity, and disease management. The between farm transmission of PRRSv is attributed to the transportation of infected animals [3], contaminated fomites, and motor vehicles [4,5], insects [6,7], and aerosols [8,9]. Among these routes, assessing the windborne local area transmission of aerosolized particles containing viable PRRSv in a near real-time manner, and allotting the level of risk has been challenging to the swine industry [10].

Infected pigs produce virus-laden aerosols when breathing, sneezing or coughing. Additionally, in animal disease settings, dried feces, dust, feed, and debris including hair containing the pathogen may also contribute as aerosolized particles containing the virus [11,12]. Airborne transmission of pathogens occurs both directly through inhaled aerosols and through contaminated objects where particles have settled [13]. Meteorological and environmental factors including directional winds of low velocity with sporadic gusts, low temperatures, high relative humidity, and low sunlight levels are suggested to influence the survival of PRRSv in air [4,14,15,16,17]. Therefore, when modeling the windborne dispersion of the virus-containing aerosols, considering virological, epidemiological, and meteorological factors is critical.

The extent to which the aerosol route is responsible for the farm-to-farm transmission of PRRSv is a long-standing controversy [10]. Experimental and observational studies have demonstrated the potential of airborne transmission of PRRSv within farms, i.e., between pens and buildings [9,18,19,20,21,22,23]. Research suggesting the co-circulation of diverse lineages of PRRSv in air samples collected around commercial sow farms in swine-dense regions further raised alarm about the risk of airborne introduction and re-introduction of the virus [24]. The between-farm transmission of PRRSv, based on observational data, has suggested the possibility of local windborne transmission up to 9.2 km [8,9]. However, there is also evidence that does not support the windborne local area transmission of PRRSv [3,17,25,26,27]. These inconsistencies led to the conclusion that the windborne local area transmission of PRRSv is an infrequent event. The potential windborne introduction is often listed as an alternative explanation when the outbreak investigation is inconclusive [10,25]. 

Estimating farm-to-farm transmission of PRRSv via the aerosol route is extremely difficult in an endemic situation, as it is often not possible to know all the emitting farms and when exactly a non-infected farm became infected. To this end, this work benefitted from the PRRSv epidemiological data available from the Morrison Swine Health Monitoring Project (MSHMP) of the University of Minnesota, which is a voluntary reporting program established to monitor important diseases affecting the U.S. swine industry [28,29] https://vetmed.umn.edu/centers-programs/swine-program/outreach-leman-mshmp/mshmp; accessed on 20 June 2021. Additionally, there are multiple steps to modeling windborne transmission, from choosing a modelling platform, determining the applicable parameter values, choosing the deposition metric to use, and defining the threshold for infection. In an attempt to address these difficulties, a recent study on PRRSv used windroses to denote the predominant wind direction between farms [27]. However, windroses fail to account for meteorological, geographical, or environmental characteristics that may lead to the survival and infectivity of the virus. 

We took an interdisciplinary approach to estimate the risk of windborne local transmission of PRRSv in a near real-time manner by using an atmospheric dispersion model (ADM). ADMs are commonly used to predict the concentration and dispersion of air pollutants emitted from sources such as power plants. Simply stated, ADMs are algorithms that predict the downwind deposition of aerosolized particles when a given quantity of substance is released into the air. Computer models that run these ADM algorithms simulate atmospheric dispersion, as well as the chemical and physical processes of aerosolized particles and gases (i.e., plume), to calculate aerosolized particles deposited at various downwind locations [30,31]. At a minimum, ADMs require inputs of the quantity of the substance released, the release location, and weather conditions. There are several types of ADMs including Gaussian plume models and Lagrangian/Eulerian Models [30]. ADMs have been used to model the long-distance dispersal of bio-aerosols, insects, and viruses that are pathogenic to humans and livestock [31,32,33,34,35]. For example, surveillance for farm-to-farm transmission of viruses via wind dispersion is mostly used for exotic animal diseases such as foot-and-mouth disease virus (FMDv) [36]. However, ADMs have never been applied to investigate the farm-to-farm transmission of PRRSv. We hypothesize that the local windborne spread of PRRSv is semi-quantifiable in relation to the predisposing meteorological and seasonal factors.

The objective of this study is to adapt an ADM to simulate the windborne local transmission of PRRSv, identify key parameters, and investigate the proportion of farm-to-farm transmission that is attributable to the windborne route using a case study. The wider objective of the study is to inform optimum time intervals for air sampling, assist in ruling out windborne dispersion in outbreak investigations, and plan preventive measures. We propose that the process of identifying the parameters and the reasoning discussed here could be useful in modeling between farm windborne transmissions of other comparable respiratory pathogenic viruses. 

## 2. Data and Methods

### 2.1. ADM Modelling Platform

A popular ADM called Hybrid Single-Particle Lagrangian Integrated Trajectory (HYSPLIT) was used to model windborne PRRSv [37,38,39,40,41,42]. HYSPLIT was developed jointly by the Air Research Laboratory of the National Oceanic and Atmospheric Administration (NOAA: https://www.noaa.gov, assessed on 1 January 2019, College Park, MD, USA), and the Australian Bureau of Meteorology. The HYSPLIT model is available free to registered and non-registered users through the NOAA Air Resource Laboratory (https://www.ready.noaa.gov/HYSPLIT.php (accessed on 20 November 2019)) in web, desktop, or LINUX—based formats. User contributed HYSPLIT add-ons are also available for platforms such as R statistical software. However, the successful use of HYSPLIT in these versions requires the careful selection, download, and storage of large files of meteorological data and often requires additional steps to run the models iteratively. Consequently, instead of using the graphical user interface (GUI) of HYSPLIT, we used the web-based API of a graphical user interface named ‘Tool for Assessing Pest and Pathogen Aerial Spread’ (TAPPAS; https://research.csiro.au/tappas/, accessed on 1 December 2018) developed by the Australian Commonwealth Scientific and Industrial Research Organisation (CSIRO) [43]. 

TAPPAS is specifically designed for long-distance wind dispersion (LDWD) of particles and has been used to model the airborne transmission of vectors of Bluetongue viral infection and African Horse Sickness [44]. HYSPLIT version 5.0.1, which is the core ADM used in the TAPPAS platform, computes air parcel trajectories, the dispersion of atmospheric particles, concentrations of particles at varying levels above ground, and particle deposition at ground level. The TAPPAS API incorporates the meteorological data as an integral component of the modeling platform and is readily usable without having to download wind data separately. The key outputs of the models are quantitative maps of the dispersed particle on the terrain over a specific period. The usual HYSPLIT deposition output results in the mass of gases and particles (i.e., plume) per square meter. The daily and cumulative deposition values over the 14-day period were regarded as alternative approximate risk values indicating the likelihood of the introduction of the virus into susceptible farms.

### 2.2. HYSPLIT-TAPPAS ADM Applicable Parameter Values for PRRSv

A literature review was conducted to determine the parameter values of airborne PRRSv applicable to HYSPLIT models [37,38,39,40,41,42]. The literature search captured peer-reviewed publications from the Web of Science, PubAg, Scopus, and JSTOR databases over the 1980–2020 period. The search terms included ‘Porcine reproductive and respiratory syndrome’ AND ‘airborne’. Two recent reviews by Anderson et al., [45] and Arruda et. al., [10], which identified the existing knowledge and gaps related to the aerosol transmission of PRRSv, were used to define the terminology used here. In this semi-systematic review, upon removal of irrelevant, duplicated, non-English, or no full-text available publications based on title and abstracts, the final selection of publications was revised. The specific objective of the review was to identify experimental or observational studies that have quantified and described values relevant to the HYSPLIT ADM parameters related to specific terms (*n* = 13: incubation period, diameter, density, temperature, humidity, speed, velocity, radiation, UV, decay, half-life, lifespan, and survival time). 

The literature search was supplemented with research on airborne respiratory diseases affecting swine populations including swine influenza and FMDv [32,33,46,47]. Additionally, a questionnaire accompanied by an interview was used to gather opinions from four experts on the modeling approach and the parameters. The experts, who represented both academia (*n* = 3) and industry (*n* = 1) were identified based on snowball sampling, and have been involved in research on the airborne transmission of swine pathogens including PRRSv and Avian influenza for over 10 years. The questionnaire included 10 open-ended questions related to the following:The type of aerosolized particles released from barns, which are used to determine the particle size.Time of the day, which was used to determine the release time of particles in the model runs.Barn architecture and ventilator heights, which were used to determine the release heights.Environmental conditions that are known to shape wind around farmsThe incubation period, time to detection of clinical signs, and general practices of conducting diagnostic tests.Biosecurity measures specific to PRRSv prevention including air filtration.Known scenarios of re-infection with the same virus variant.Any changes in on-farm activities once a herd is detected with an outbreak.Observed or known seasonal and geographical characteristics of PRRSv transmission.Any comments on the windborne between farm transmission of PRRSv.

The parameter values involved epidemiological features of the virus and the survival of the aerosolized virus in relation to key meteorological features [43,44]. Specifically, these included aerosol particle diameter, density, release height and quantity, estimates for the maximum time the virus could remain infective in the air, and the estimated decay. The relevant meteorological parameters included wind direction, speed, turbulence, temperature, and relative humidity. Given that the models were run using the TAPPAS Web API, parameters used in the TAPPAS runs were targeted in the parameter identification process. It is important to note that while parameters were collected for the HYSPLIT models, the current version of the TAPPAS platform was not enabled to intake temperature, humidity and UV tolerance levels of the virus; therefore, these parameters were not used in the sensitivity analysis. A dispersal window of two weeks, during which the incursion event i.e., as a result of receiving a sufficient amount of aerosolized particles containing the virus and therefore an outbreak in a susceptible farm was most likely to have occurred, was assumed. Further details on the choice of two-week period is presented under the case study and the justification of parameters. 

The HYSPLIT model for particle dispersion used in TAPPAS allows for the transport of particles with mean wind and a random component to account for turbulence [34,43,44]. The maximum altitude was set to 10,000 m above the ground level (m-AGL), i.e., once the top of the model is defined, the aerosolized particles that reach the top are reflected, i.e., bounced back into the model during the simulations. Both ‘dry’ (gravitational) and ‘wet’ (rainfall) deposition were permitted. These deposition parameters include velocity, average weight, A-Ratio, D-Ratio, effective Henry’s constant, in-cloud, and below-cloud [43,44]. Further details on Lagrangian models [48], dry deposition of airborne viruses [49], and details on the dry and wet deposition of atmospheric gases and modeling dry and wet deposition on HYSPLIT are found elsewhere [38,42]. The limitations of HYSPLIT and similar ADMs are discussed elsewhere [30,31].

### 2.3. Case Study

#### 2.3.1. Disease Data

To demonstrate the utility of the identified parameters on HYSPLIT-TAPPAS ADM, a case study was conducted. We used the MSHMP database of the University of Minnesota [28,29], https://vetmed.umn.edu/centers-programs/swine-program/outreach-leman-mshmp/mshmp, accessed on 20 March 2020, which includes the weekly PRRSv infection status of swine farms and the farm characteristics including the geolocation, the number of animals, vaccine status, and farm air filtration status. The participants submit their status to the MSHMP program on a weekly basis. Participants voluntarily report whether there has been a change in their status: whether there has been an outbreak based on clinical signs together with PCR positive and sequencing or whether the herd has improved its status from producing PCR positive pigs (i.e., unstable status) to producing PCR negative pigs (i.e., stable status). Once the herd reaches stable status, the herd can continue to either vaccinate sows and gilts, acclimate gilts with live virus inoculation, or work towards full elimination. These latter statuses are known as positive stable with live virus usage, positive stable with field virus, positive stable with no vaccination or acclimation, undergoing elimination with recently introduced seronegative gilts, or fully negative. Here we classify the positive unstable herds as the ‘emitting’ farms. The producer companies conduct diagnostic testing based on clinical signs unless the testing was initiated due to PRRSv infected farms in close proximity or farms where animals were received within recent weeks. For the simplicity of the analysis, the disease status was dichotomized where farms with a positive diagnostic test were regarded as the infectious i.e., ‘source’ and all other statuses were considered as ‘susceptible’, regardless of the vaccine status. These infectious farms were considered as emitting locations generating aerosols containing the pathogen (PRRSv). 

The case study included PRRSv statuses of the MSHMP farms in Minnesota. The data included 167 swine farms in Minnesota including commercial (54%), farrow-to-wean (20%), farrow-to-finish (15%), multiplier (6%), boar stud (1%), sow (2%), and other (2%). The inventory of pigs ranged from 30 to 6000 with a mean of 1975 heads. Cases from 29 March–12 April of 2017 were chosen for the case study. This particular time point was chosen because compared to the number of ongoing outbreaks on 29 March 2017 (*n* = 29) and 05 April 2017 (*n* = 29), six new farms were reported with new outbreaks on the following week of 12 April 2017 (*n* = 6) (Table 1). We define an outbreak as farms that have been diagnosed as positive for PRRSv and are experiencing and reporting an ongoing status of the disease on the premises in the relevant week, regardless of the PRRSv lineage. Given that the farms may enter, exit, or not submit data for certain weeks, the total number of farms in the monitoring for each week is expected to vary. The farms that were non-infected as of 12 April 2017 were considered the susceptible farms for that week. One of these six farms was installed with air filters in 2009, whereas the filtration status of the others was unknown. Forward dispersion model runs were performed using the TAPPAS Web API with direct access to HYSPLIT running on a high-performance cloud computing server. The objective of the forward runs was to assign the risk of PRRSv introduction based on the deposition of particles on all farms. Three metrics were used to assess the potential amount of virus being deposited on susceptible farms, *viz;* the cumulative 14-day deposition, the median daily deposition, and the maximum daily deposition.

#### 2.3.2. Wind Data

The meteorological fields for wind were obtained from the National Oceanic and Atmospheric Administration’s gridded meteorological database (https://www.ready.noaa.gov/archives.php, accessed on 20 January 2020). Specifically, the North American Mesoscale Forecast System (NAM) Hybrid sigma-pressure archive (CONUS, Alaska, Hawaii, 2010-) (ftp://arlftp.arlhq.noaa.gov/pub/archives/nam12, accessed on 1 April 2020), which includes data for several weather parameters including temperature, precipitation, lightening, and turbulent kinetic energy (i.e., wind) was used. NAM contains wind data from 2010 onward, at 12 km horizontal resolution and 3-h temporal resolution. NAM data is incorporated as an integral part of the TAPPAS platform enabling the retrospective and prospective analysis of the windborne dispersion of particles. In the model runs, the vertical velocity of the particles was defined by meteorological data. For the simplicity of the analysis, the emitting farms were considered to excrete the same amount of the virus every hour throughout the model run time. The particles were transported with the mean wind plus a random component of motion to account for atmospheric turbulence making the cluster of particles expand in time and space. In the TAPPAS simulations, particles were considered to be infectious and to contribute to airborne transmission regardless of the temperature, humidity, and other relevant meteorological parameters. All other settings such as horizontal and vertical mixing coefficients were used at the default settings of HYSPLIT. 

#### 2.3.3. Sensitivity Analysis 

To determine the key parameters that the HYSPLIT-TAPPAS model is sensitive to, a sensitivity analysis was conducted with independently varying input parameter values for: (1) model duration (h), (2) takeoff/emission height (m-AGL), and (3) release quantity, i.e., the particle released from emitting farms. Over weeks two and three of the case study period, a total of 252 simulations were run from the 29 emitting farms, corresponding to 18 variable combinations (three release quantities (100, 1000, and 10,000); two release heights (0.5 m and 4 m); and three model run periods (12, 24 and 36 h) over the 14 days. The assigned number of particles emitted is a semi-quantitative representation of low (100), medium (1000), and high (10,000) number of particles. This created 18 unique scenarios (i.e., 3 × 2 × 3 = 18). The range of deposition values resulting from all 18 scenarios on the study farms are summarized in Appendix A. From these runs, we estimated the three-deposition metrics for each of the 131 susceptible farms on the week of 12 April 2017, providing 34,020 deposition data points. The three metrics included the cumulative 14-day deposition at each of the farm locations, the median daily deposition, and the maximum daily deposition. To determine a threshold deposition for infection, we presumed that farm-to-farm aerosol infection is uncommon and that only the higher deposition values might result in a farm becoming infected by this route. To estimate this, we produced histograms of the deposition values for the three metrics and applied natural breaks/Jenks classification [50] for the cumulative, median, and maximum daily deposition. The frequency distributions were divided into three classes using two natural breaks (i.e., Jenks).

The model outputs were maps depicting the plume deposition per square area (mass/m^2^), i.e., the ‘foot print’ of aerosol deposition. The maximum particle deposited at each of the susceptible farm locations was extracted by intersecting the farm locations with the output maps. The particles deposited at each susceptible farm during the two-week period were used to represent the “exposure hazard” (i.e., potential for pathogen introduction via aerosols). In this retrospective modeling exercise, it is challenging to determine an infectious dose that is sufficient to cause the windborne disease. 

We presumed that a very low exposure dose is unlikely to result in infection, and to convert the exposure hazard into a “potential risk of infection”, we applied an infecting dose threshold [50]. As discussed further under parameter values resulting from the literature review, the infectious dose for aerosol exposure under experimental conditions can range from as low as 2 TCID_50_ up to 103.1 TCID_50_ [16,51]. In the absence of field data relevant to the farms in the study, we determined these thresholds by examining the histograms of the cumulative, median daily, and maximum daily concentrations. Applying these thresholds to the susceptible farms, we thus were able to classify them as receiving a potentially sufficient infecting dose or not; the same farm that was classified as a PIF under multiple sensitivity scenarios was referred to as a “potentially infected farm” or “PIFs”. By estimating the average effect of the values of the three modelling variables, these being the release quantity, model run time, and release height on the number of PIFs, we were thus able to determine how sensitive the model was to these parameter values. The model sensitivities to each of the variables were summarized in tornado plots. The baseline of the tornado plot is calculated by taking the average number of PIFs for each of the 18 scenarios that are above the threshold. Additionally, the distance to the PIFs from the closest emitting farm that is upwind were calculated. In this distance to PIF calculation, a farm may be categorized as a PIF under the threshold for cumulative deposition but may not be categorized as a PIF under the median daily deposition. Therefore, all farms under all simulation scenarios were considered when calculating the distances. 

## 3. Results

### 3.1. Parameter Values and Rationale

The literature search resulted in 115 unique publications; upon revision, 90 publications were used to identify PRRSv parameters relevant to Table 2. The full list of references included in the review is included as a Appendix A. Given the limited number of data points available to support ADM applicable parameter values, no meta-analysis was incorporated. Other airborne respiratory diseases affecting swine populations and expert opinion supplemented the literature search. The HYSPLIT equivalent parameter names and the selected parameter values used in the sensitivity analysis on TAPPAS (i.e., HYSPLIT-TAPPPAS) are listed in Table 3. 

The time lag of between-farm transmission of 14-days was assumed based on the estimated PRRSv incubation period or generation time [52,53]. Animals exposed to PRRSv incubate the virus for an estimated period of 14 days, and these infected animals are capable of spreading infection between 3–40 days post-infection regardless of the appearance of clinical signs [53]. Therefore, the duration of emission of the infective particles was assumed to be 14 days. Published work suggests that airborne transmissibility of PRRSv was lineage-dependent. While several lineage s of PRRSv tend to co-circulate in areas with high swine farm densities [24], the temporal dynamics in the field suggest that only two to three viral sub-lineages s are dominant at any given time [54]. Yet, as an exploratory attempt to model PRRSv using ADMs, this work contributes to the literature by identifying the key parameters. 

In general, respiratory particles produced during breathing, sneezing, or coughing vary in their size. Those that are >5–10 μm in diameter are referred to as ‘respiratory droplets’, and tend to travel shorter distances in the air [55,56,57,58,59]. Whereas respiratory particles that are <5 μm, referred to as ‘droplet nuclei’, can be transported via air for >1 m distance [57]. The viral particles that are entrapped and remain infective in the droplet nuclei, i.e. bioaerosols, are carried away with wind contributing to windborne route of pathogen transmission [60]. Given that the naturally produced aerosols by pigs infected with PRRSv in an experimental setting varied between 0.4–10 μm in diameter [61], an assumption was made that the average aerosol diameter was 5 μm. Alonso et al.’s study further suggested that the PRRSv viral quantities in the aerosols ranged between 6 × 10^2^ (in aerosol diameter 0.4–0.7 μm) to 5.1 × 10^4^ RNA copies/m^3^ (9.0–10.0 μm). In experimental conditions, the density of respiratory nuclei was described to have an average value of 0.7 g/cubic centimeters [61,62]. In another publication, the buoyant density of infectious PRRSv viral particles was estimated to have 1.18–1.22 g/cc [63]. 

Each virus reacts uniquely to each or a combination of factors, depending on the structural composition of the virus and its interactions with other components of the aerosols [64]. Being an enveloped RNA virus, PRRSv survivability outside of the host is affected by temperature, pH, and exposure to detergents [62]. It has been reported that aerosol transmission of PRRSv depends on physical variables related to the infectious particles such as particle size [61], quantities of pathogens emitted, the rate of droplet desiccation, and environmental factors such as temperature and relative humidity [13,64]. It is known that PRRSv can survive for extended intervals (>4 months) at temperatures ranging from −70 to −20 °C [62]; however, viability decreases with increasing temperature. Specifically, recovery of PRRSv has been reported for up to 20 min at 56 °C, 24 h at 37 °C, and six days at 21 °C [62]. PRRSv requires low relative humidity for the optimal preservation of infectivity, i.e., <30% relative humidity [14,64]. There is limited data available on the PRRSv sensitivity to UV radiation. An experimental setting that used a lamp emitted UVC radiation at 253.7 nm reported no effect of UV radiation in reducing aerosolized PRRSv [65]; however, this was also attributed to the insufficient contact time and the required exposure times were unclear at the time of the study. Another study concluded that an average UVC intensity of 1,1170 μW/cm^2^ at 19% relative humidity could reduce 99% of aerosolized PRRSv that was aerosolized from a culture suspension of 10^6.1^ TCID_50_/mL in an experimental setting [66]. A recent model-based experimental study by Li et al., [67] suggested that UV doses needed to inactivate 3-log (i.e., 99% reduction) of aerosolized PRRSv with UV 254 nm dose were 0.521 and 0.0943 MJ/cm^2^, based on one-stage and two-stage models, respectively. 

In the absence of field studies, several parameter values were extracted from experimental study settings. For example, Alonso et al., [68] demonstrated that retrograde air movement at the minimum velocity of air moved at 0.76 m/s was sufficient to introduce aerosolized PRRSv into filtered air spaces. Due to the risk of windborne transmission of several pathogens including PRRSv, the filtering of incoming air to pig facilities in swine dense regions has been proposed as a preventive measure [69,70]. In compatible HYSPLIT models for FMDv, the dry deposition velocity that would be applicable for PRRSv is 0.01 m/s [33]. However, in HYSPLIT models, assigning velocity values overrides the gravitational settling calculations of particles based on their size parameters; therefore, only particle diameter, density and shape were used in the models.

In the airborne modeling of FMDv, a decaying constant is commonly used to simulate the biological aging of the virus [32]. In this assumption, the decay of the virus was assumed to be exponential as in the case of radioactivity [32]. Previous studies on FMDv suggested decaying times of 30 min [33] and 2 hr [32,71], respectively. This constant was also said to depend on the strain of the virus. Similarly, the survival of PRRSv depends on temperature and humidity [14,16]. PRRSv has been experimentally shown to survive between −70 through 30 °C and a relative humidity of 25% through >80%, i.e., low temperatures and high humidity may act in favor of the survival of the virus [16]. The particle numbers emitted from a source location are meant to represent aerosol emission and the risk of deposition is meant to represent the particles that would be sufficient to cause the disease in a susceptible population. In the HYSPLIT model developed and validated for FMDv [71], the virus emission rate was set as TCID_50_ units per hour. Translating the quantity of computer model particles deposited to the empirical infectious dose of PRRSv that would cause the disease is beyond the scope of this study. The key HYSPLIT input parameters relevant to particle dispersion include emission height, emission quantity, and the model runtimes. Hence, these parameters were subjected to sensitivity analysis. The emission heights of 0.5 m and 4 m were used to represent the height of exhaust fans and the attic of the swine barns [72]. 

**Table 2 viruses-14-01658-t002:** Epidemiological and virological parameters of aerosolized PRRSv as modelled using HYSPLIT [38,39,40,41,42].

Parameters	Search Terms	Values	References
Time lag for between farm transmission	incubation period	14 days (Assumption)	[52,53]
Particle diameter (aerosolized particle diameter)	diameter	5 μm (0.4–10 μm)	[61]
Particle density	density	0.7 g/cc	[61]
Minimum temperature deg C	temperature	−70	[63]
Maximum temperature deg C		30	[14]
Minimum relative humidity %	humidity	50% (25–79%)	[14,16]
Maximum relative humidity %		100% (≥80%)	[16]
Minimum wind speed m/s	speed, velocity	0.76 m/s (In-door experimental settings)	[68]
Maximum wind speed m/s		0.01 m/s (dry deposition velocity)	FMDv: [33]
Maximum UV radiation MJ m^−2^	radiation, UV	5210 MJ/m^2^ (In-door experimental settings)	[67]
Exponential decay constant	decay	1.0 × 10^−4^/ second (Decay constant λ = 6.4 × 10^−4^)	FMDv: [32]
	half life	4.1 min (At 30 deg C and 50% relative humidity)	[14]
		120 min virus half-life	FMDv: [71]
Lifespan	lifespan	3 days (Varies with temperature and humidity)	
Maximum time in air (Alive & infective)	survival time	1 h–4 weeks (Varies with temperature and humidity)	[5,8,69]

**Table 3 viruses-14-01658-t003:** The Tool for Assessing Pest and Pathogen Aerial Spread (TAPPAS) [43]; https://research.csiro.au/tappas/ (accessed on 20 November 2021)) Web API input parameters, equivalent variables for the web version of Hybrid Single-Particle Lagrangian Integrated Trajectory (HYSPLIT) [41], and the parameters used when simulating Porcine Reproductive and Respiratory virus (PRRSv) containing aerosols on TAPPAS. The three parameters and relevant values assigned to conduct sensitivity analysis of PRRAv aerosols are highlighted in grey: (a) model duration, (b) takeoff height, and (c) particle concentration at emitting farms.

TAPPAS Web API Input	HYSPLIT READY * Web Application Equivalent Variable Name	Parameter Value Setting for the PRRSv TAPPAS Runs	Justification or Explanatory Note
Location	Source location: latitude, longitude	Coordinates of outbreak farms (latitude, longitude)	
Species	Release type	New Species: PRRS	Maximum of 4 characters for HYSPLIT input names
Meteorology	Meteorology	NAM 12 km (hybrid sigma-pressure)	
Vertical Motion: Default setting	Model vertical velocity	0	Using the vertical velocity fields within the meteorological data.
Output: Concentration, deposition	Output	Deposition (mass/m^2^)	
**Source Term Parameters**			
Direction: Forward, backward	Dispersion direction	Forward	
Date of run	Release start time: year month day hour minute	Everyday	User defined
Take off/release start time		12 am	User defined
Release height(s)	Release top	0 and 4 m-AGL	User defined: Sensitivity analysis—
	Release bottom	-	Feature not implemented in TAPPAS
Release quantity (mass/hr)	Numpar—Limit of the number of computing particles released per time period	−100, −1000, −10,000 mass	User defined: Sensitivity analysis Note: Specifying a negative value ensures a constant particle release per hour for each source location.
Maximum time in air	Khmax—Release duration: hours minutes	72 h	Presumed maximum hours of infectivity of PPRSv
Maximum release quantity	Maxpar—Limit of the total number of computing particles tracked at a time	Varies with each run based on the number of emitting particles	
**Runtime Parameters**			
Model run time (From 1st release)	Total run time (hours)	12 h, 24 h, 36 h	User defined: Sensitivity analysis
Release duration	Averaging period/output interval	24 h	
Run type (single day or multiple days specified by user)	Single runs only	Single day	
Top of averaged layer—default setting	Top of averaged layer	100	
**Deposition Parameters**			
Particle characteristic	Pollutant characteristics: particle, gas	Integral: particle	
Particle diameter, Density, Shape	Particle diameter (μm), Density (g/cc), Shape	5, 0.7, 1	
**Dry Deposition:** Velocity (m/s), Molecular Weight (g), A-Ratio, D-Ratio, Effective Henry’s Constant	Velocity (m/s), Molecular Weight (g), Surface Reactivity Ratio, Diffusivity Ratio, Effective Henry’s Constant	0, 0, 0, 0, 0	
**Wet Deposition:** Actual Henry’s constant, In-cloud (L/L), Below-cloud (1/s)	Actual Henry’s constant, In-cloud (L/L), Below-cloud (1/s)	0, 8.0 × 10^−5^ L/L, 8.0 × 10^−5^ L/L	Default of HYSPLIT;
Exponential decay constant (λ)	Radioactive decay, i.e., virus half-life (days)	0	Default value of 0 set within TAPPAS
**Sampling**			
Sampling type	Sampling interval: type hour minute (0 = Average, 1 = snapshot, 2 = maximum)	Average concentrations	User defined
Sampling period/interval	Sampling interval: type hour minute	12 h, 24 h, 36 h	User defined: Sensitivity analysis
Height of model (m-AGL)	Height of each level (m)	0 m-AGL	Level of output—User defined: 0 = deposition (mass/m^2^) and >1 = plume concentration (mass/m^3^)
Top of model (m-AGL)	Top of model domain (internal coordinates m-agl)	10,000 m-AGL	User defined
**Display Options**			
Model sampling/output grid cell resolution	Grid spacing (deg) Latitude, Longitude	0.012	User defined
Window size N/S location centroid	Grid span (deg) Latitude, Longitude	25 degrees	User defined
Window size E/W location centroid		25 degrees	User defined
Output	GIS output of contours	Google Earth (Kmz)	Default of TAPPAS

* The online version of HYSPLIT on READY platform was used to define the HYSPLIT parameter names (https://www.ready.noaa.gov/HYSPLIT_disp.php, accessed on 20 January 2020); [41]. Results of the case study.

The infectious dose of the virus depends on the transmission route as studied for PRRSv [14,73] and FMDv [74]. In an experimental setting, aerosolized exposure doses of 10^4.6^–10^5^ genomic copies per ml (i.e., 10 2.7 to 10 3.5 TCID_50_ per ml) were sufficient to infect >90% of the exposed pigs [51]. Another dose-response modeling study published in 2021 suggested that PRRSv was most infectious via the aerosol route compared to the intranasal and oral routes of infection [73]. PRRSv is highly infectious, with 10 or fewer virions being sufficient to experimentally infect pigs when administered intra-nasally or intra-muscularly [75]. The infectivity of PRRSv is known to vary with the strain and the route of infection [76,77]. Cutler et al., [16] calculated that the infectious dose (ID_50_) for the aerosol exposure to isolate MN-184 was less than 2 TCID_50_, while [51] reported an ID50 of 103.1 TCID_50_ for the aerosol exposure using isolate VR-2332. 

### 3.2. Deposition Thresholds

After 14-days of deposition under each scenario, applying natural break thresholds to the three deposition metrics (i.e., 14-day cumulative, median, and maximum daily deposition) enabled the identification of potentially infected farms (PIFS). In this semi-quantitative assessment of windborne particle deposition, exposed farms that received a deposition greater than the first threshold value were considered and hence were classified as PIFS (i.e., farms that have received a sufficient number of particles to be classified as at risk for airborne introduction of the pathogen).
The cumulative 14-day deposition resulted in two natural breaks at 3.13 × 10^−5^ and 1.4 × 10^−4^ mass/m^2^. At 1.4 × 10^−4^ mass/m^2^ there were *n* = 50 PIFs for all the 18 scenarios, representing *n* = 13 unique farms (Figure 1: Panel A).The median daily deposition resulted in two natural breaks at 1.0 × 10^−6^ and 5.50 × 10^−6^ mass/m^2^. At 5.5 × 10^−6^ mass/m^2^ there were *n* = 119 PIFs representing *n* = 41 unique farms (Figure 1: Panel B). The farms classified as PIFs under the cumulative 14-day were also included within the median daily deposition categorizations.The maximum daily deposition the susceptible farms received belonged to three maximum daily deposition categories. The two natural breaks were at 1.0 × 10^−6^ and 1.0 × 10^−5^ mass/m^2^. At 1.0 × 10^−5^ mass/m^2^ there were *n* = 596 PIFs for all scenarios representing *n* = 105 unique farms (Figure 1: Panel C).

Due to the grouping of the maximum deposition into a limited number of values (i.e., 8) the Jenks method was less successful in defining the upper threshold. Therefore, we manually adjusted the threshold to 1.50 × 10^−5^, which was a comparative position in the distribution to that of the median and cumulative histograms, and thereby gave *n* = 51 PIFs for all scenarios representing 15 unique farms.

### 3.3. Six New Cases on 12 April 2017

In general, the plume densities deposited on the study terrain, as captured by the deposition footprint at every 12th, 24th or 36th hour, ranged between six levels: 0 (10^−9^ mass/m^2^), 1 (10^−8^ mass/m^2^), 2 (10^−7^ mass/m^2^), 3 (10^−6^ mass/m^2^), 4 (10^−5^ mass/m^2^), and 5 (the maximum concentration possible) (Figure 2). In the case study where emissions from 29 March 2017–12 April 2017 period were used to predict the depositions on susceptible farms, all six farms reported a new outbreak on 12 April 2017 had received wind from an emitting farm at least once during the 14-day period. Among these six farms, five were clustered in the southeast of the state and were away from the majority of the emitting farms (>57 km away from any of the emitting farms). Only one was in the south of the state and was close to other emitting farms (i.e., within <10 km of emitting farms) (Figure 2). Among the six newly infected farms, this one in the south of the state is the only one qualified to be categorized as a PIF under two of the 18 scenarios. The two scenarios of 36 h model run time with 10,000 particles released at heights of 0.5 and 4 m AGL resulted in a cumulative deposition ≥ 1.4 × 10^−4^ mass/m^2^, a median daily deposition ≥ 5.50 × 10^−6^ mass/m^2^, and a maximum daily deposition ≥ 1.0 × 10^−5^ mass/m^2^. 

### 3.4. Sensitivity Analysis

Among the three parameters tested for model sensitivity, the TAPPAS-HYSPLIT ADM was most sensitive to the number of particles emitted, followed by the model run time, and the release height was the least sensitive among the parameters tested. As summarized in the tornado plots, the baseline (i.e., the average number of PIFs for each of the 18 scenarios that are above the threshold of 1.4 × 10^−4^ mass/m^2^ for the 14-day deposition, was 2.78 farms. The baseline for the median daily deposition was 6.61 while the baseline for maximum daily deposition was 2.83 PIFs (Figure 3). All PIFs were within 25 km of the closest emitting farm (Figure 4), with 53.66% within 10 km.

## 4. Discussion

Quantifying the extent to which the aerosol route is responsible for the farm-to-farm transmission of PRRSv, and allotting the level of risk, especially in near real-time is an ongoing challenge. This work gathers relevant input parameter values using published literature and expert opinion, and successfully exemplifies the adaptability of NOAA-HYSPLIT to model windborne dispersion of aerosolized PRRSv between farms. In this semi-quantitative study, when the determined parameters and assumptions were used to simulate the windborne transmission of PRRSv, those receiving a median daily deposition of ≥5.50 × 10^−6^ mass/m^2^ and are within 25 km of an emitting farm were considered to be at potentially high-risk. However, it is important to note that atmospheric dispersion modelling by itself cannot definitively prove that aerosol transmission was responsible for a new outbreak on a farm and must be interpreted in the context of the study. Using an ADM to show that a high concentration of the virus might have been deposited only establishes windborne transmission as a plausible pathway for the introduction of a pathogen onto a farm. Dispersion modelling is useful in “ruling out” farm-to-farm aerosol dispersion as a transmission route when the results show that deposition is not possible or at a low concentration [44]. For example, in a farm-level outbreak investigation, which explores the role of multiple routes of pathogen introduction such as animal movements and fomites, this modeling approach may complement ruling out the airborne introduction at the very least. We propose that our study would enable an improved outbreak investigation to answer the question “what proportion of farms are at high risk for PRRSv transmission caused by aerosolized particles?” for the US swine industry.

In the absence of near real-time windborne models to be used in outbreak investigations, a distance band of 5 km or 10 km radius from the infected farm is often used as a proxy for airborne transmission in PRRSv outbreak investigations and epidemiological models [3,27]. The case study here demonstrated assigning up to a 25 km radius from an emitting farm when the median daily dispersion is considered (Figure 4). However, when compared across the three matrices used, the cumulative 14-day depositions were within 10 km and 53.3% of the median daily depositions were on farms within 10 km from an emitting farm (Figure 4). The case study further demonstrated that for at least one of the six newly infected farms of the week of 12 April 2017, the windborne route was a potential pathway of farm-to-farm spread. By contrast, the potential for windborne transmission of PRRSv to the cluster of five new outbreak farms in the southeast of Minnesota can be ruled out, within the limitations of the study. PRRSv outbreaks are not reportable in the U.S. and therefore there are no regulations requiring outbreak investigations or control zone radii. The findings of the case study here indicate that windborne aerosolized PRRSv have the potential to reach farms that are beyond 10 km within a window of the 14-day shedding period given the wind direction and speed. However, it is important to recognize that the work here is model-based using a set of parameters, of which not all are validated. Moreover, the HYSPLIT-TAPPAS models used here were not informed with parameters on temperature and humidity thresholds, which determine the viability and infectivity of the virus. Therefore, 25 km PRRSV airborne transmission is suggested under the conditions of this case study. This may inform estimating the optimum time intervals for air sampling and planning preventive measures including the installation of barn air filters, UV irradiation, or electrostatic particle ionization [62,67,74,79,80,81].

According to the sensitivity analysis, the model was most sensitive to the number of particles emitted. This indicates the importance of focusing future work on estimating the virus excretion from outbreak farms. More research is needed to improve the use of ADMs to accurately quantify the risk of windborne transmission because the virus produced can be expected to vary with the herd size and intra-herd disease prevalence. Development of a fully integrated PRRSv farm-to-farm windborne dispersion risk assessment comparable to that currently available for FMDv on HYSPLIT [71] will require additional research effort, as there are still a large number of knowledge gaps related to the airborne transmission of PRRSv [10]. One approach might be to incorporate the outbreak status of farms using compartment modeling alongside HYSPLIT models, as done for FMDv [71]. The lack of model sensitivity to emission heights of 0.5 and 4 m-AGL is attributable to the NAM wind data layer characteristics in which the vertical wind data layer is consistent up to 100 m-AGL. 

A few of the limitations in the parameter determination and the case study include the patchiness of the data, disregarding the differences between PRRSv lineages, assuming a 14-day infectious period, and not incorporating temperature, humidity, and UV tolerance levels in HYSPLIT-TAPPAS model runs. While a reasonable time lag of 14 days was assumed in order to receive a sufficient amount of aerosolized particles from an emitting farm, meteorological conditions such as temperature and humidity would affect the aerosol evaporation and the ability of viruses to remain infective. Moreover, it is important to consider that this time range may vary by season [64]. For example, droplets or aerosols exhaled by animals shrink rapidly with the lower humidity outside the respiratory airway, creating smaller aerosols [64]. However, as described under methods, the HYSPLIT- TAPPAS platform was used because it is modified to model the airborne transmission of pathogens, and the meteorological data are incorporated as an integral part of the platform, which was convenient for the case study. Incorporating the meteorological parameter thresholds relevant to viral survival would not change the footprint of the particles; however, the deposition values may be subject to variation. A detailed study with years of disease data and incorporating all parameters along with model cross-validations are required to confirm the results determined here. Estimating farm-to-farm transmission in an endemic setting is difficult and with a voluntary testing program, it is not possible to know the status of all farms. PRRSv transmission dynamics exhibit viral fadeout and reintroduction, where the persistence of outbreaks are associated with larger herds in pig-dense regions with the continuous introduction of infectious stock [78]. Therefore, having data from all pig farms in a given area would improve the accuracy of model prediction and risk estimation. While patchy, with a coverage of over 60% of the farms in the state, MSHMP data provides a unique opportunity to estimate the windborne transmission of PRRSv in an endemic setting. Previous work that used HYSPLIT modelling also used the backward modeling approach, which enables the determination of the up-wind region where emitting farms might be located [44,78]. This backward modeling is an alternative approach to estimating the risk and the emitting source of infection, especially when the disease data are limited [76]. HYSPLIT models take the topography into account; therefore, to generalize the findings here to another geographical area requires further studies in different geographical areas.

The future direction of this modeling is to use the HYSPLIT model for years of disease data available from MSHMP and incorporating temperature, humidity, and UV tolerance levels; and investigating the seasonal changes of wind and its impact on deposition. Evaluating the impact of potential confounders and effect modifiers such as the number of animals, age, filtered vs. non-filtered state of the barn, and seasonality would improve the use of the approach further. For example, Linhares et al. [79] described the use of a PRRSV modified-live vaccine as a tool to reduce viral shedding to the environment, including aerosols, which suggests that underlying herd immunity plays a major role in aerosol excretion, and potential for transmission to nearby swine populations [10]. Therefore, an essential modification in the future use of ADMs is to enable the input of individual farm characteristics. Literature suggests that younger pigs are more susceptible to infection, have higher levels of viremia, and excrete virus at higher concentrations [80]. Similarly, the number of particles emitted would not stay constant; with the progression of the disease and the number of animals affected, the emission profile would change over time [71,81]. Therefore, an ideal approach would be to incorporate both a compartmental epidemiological modeling approach where the emission is modeled based on population characteristics and then use HYSPLIT for the airborne component, as was done for FMDv [33,71,81]. 

## 5. Conclusions

Atmospheric dispersion models can be successfully adapted to estimate the risk of windborne transmission of PRRSv between swine farms. When the identified PRRSv-specific parameter values were used to simulate windborne transmission using ADMs, the model was most sensitive to the number of particles emitted, followed by the model run time, and the height of emission was the least sensitive among the three parameters tested. Within the context, parameters, and assumptions used in the case study, we concluded that a farm with an ongoing outbreak would pose a non-zero risk of infecting a susceptible herd up to 25 km if the infected animals shed the virus for 14-days. Incorporating ADMs to model the windborne dispersion of pathogens in a near real-time manner as exemplified would improve outbreak investigations, planning preventive measures, and inform targeted disease management. 

## Figures and Tables

**Figure 1 viruses-14-01658-f001:**
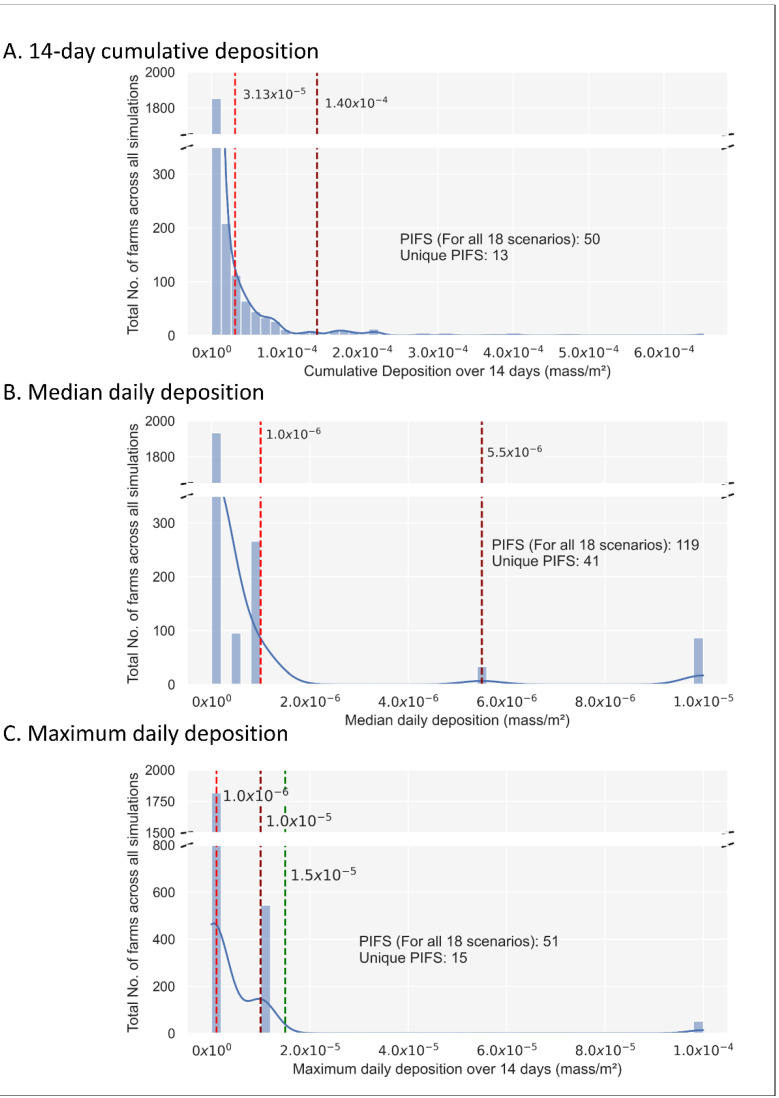
Histograms depicting the frequency distribution of the number of exposed farms under the three deposition metrics relevant to all 18 simulated scenarios (excluding the emitting farms). Dashed red lines indicate the first and second natural Jenks breaks [50], whilst the green line represents a manually defined break. Exposed farms which received a deposition greater than the second threshold value (Panels (**A**,**B**)) or the manually defined break (Panel (**C**)) were classified as potentially infected farms (PIFS). The number of Potentially Infected Farms (PIFs) for all 18 scenarios and the number of unique PIFs are listed.

**Figure 2 viruses-14-01658-f002:**
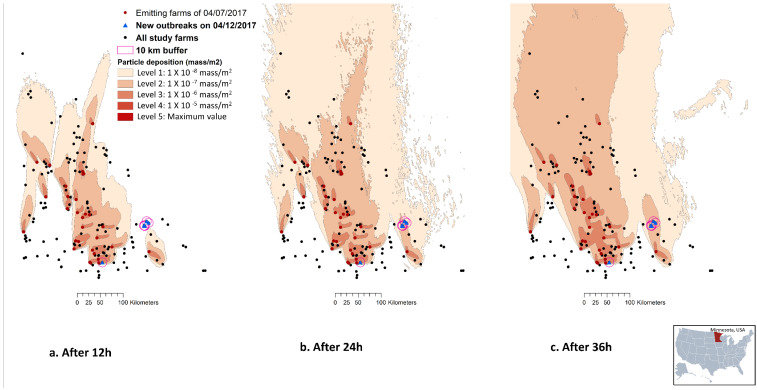
Illustration of an example HYSPLIT by NOAA: https://www.noaa.gov)—(TAPPAS; https://research.csiro.au/tappas/) output maps. The maps depict the deposition footprint of the 07 April 2017 emissions that resulted from the scenario of 10,000 emission particles at 4k m-AGL after (**a**) 12 h, (**b**) 24 h, and (**c**) 36 h. The map also illustrated all pig farms (n = 167 in Minnesota, USA) relevant to the case study from the 29 March 2017–12 April 2017 period, and the six swine farms reported a new Porcine Reproductive and Respiratory virus (PRRSv) outbreak on 12 April 2017. The wind data from the North American Mesoscale Forecast System (NAM) was used to model particle dispersion.

**Figure 3 viruses-14-01658-f003:**
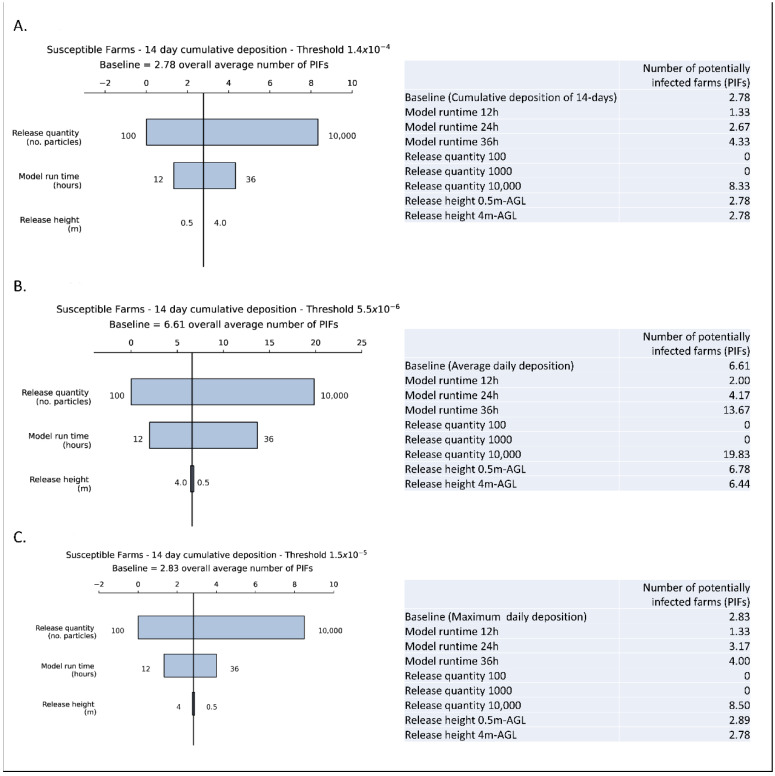
Tornado plots showing the average number of potentially infected farms (PIFs) in response to the values of the three modelled parameters, viz. release quantity (*n* = 3; 100, 1000, 10,000), model run time (*n* = 3; 12 h, 24 h, 36 h) and release height (*n* = 2; 0.4 and 4 m-AGL). Plots were calculated using the three estimates of deposition: (**A**) Cumulative 14-day, (**B**) Median daily and (**C**) Maximum daily deposition. The point at which the vertical axis crosses the horizontal axis is the PIFs for the relevant deposition measure in the baseline, while each horizontal bar represents the range of depositions for each variable, as labelled on the right hand side. For each variable, the inputs that resulted in the lowest and highest means of depositions for that variable are listed at either end of the horizontal bar.

**Figure 4 viruses-14-01658-f004:**
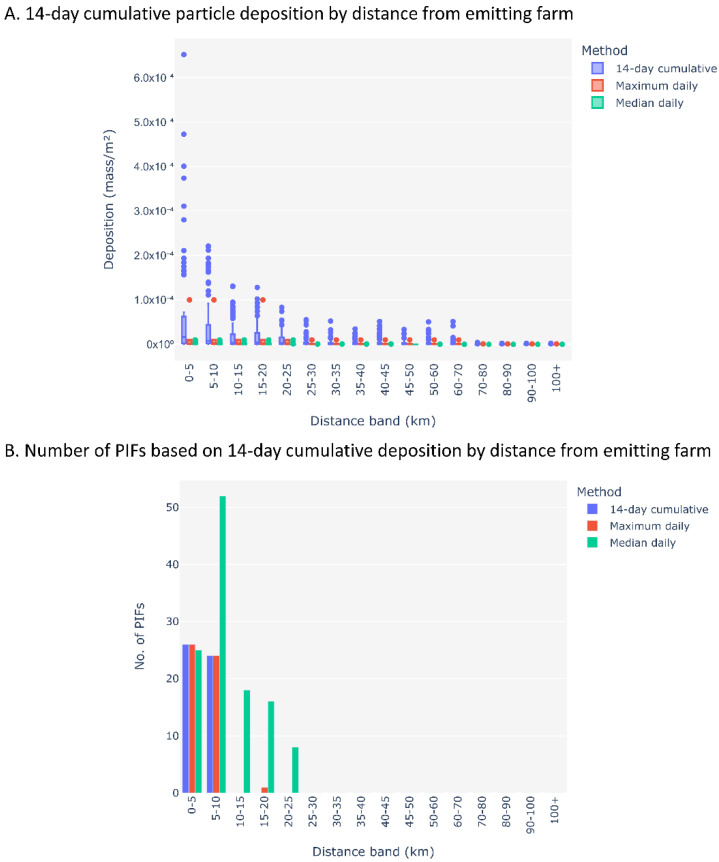
Plots depicting the distance-wise distribution of 14-day cumulative, median daily, and maximum daily depositions of aerosolized particles containing PRRSv modeled using HYSPLIT. Counts of all farms considered as potentially infected farms (PIFs) receiving sufficient deposition under each method for and each of the 18 scenarios were used.

**Table 1 viruses-14-01658-t001:** Breakdown of the case study farms subjected to atmospheric dispersion modeling (ADM) simulation.

Week	Week Start Date	Number of Participant Farms	No. Infected and Excreting Farms	No. Non-Infected Susceptible Farm Per Week	Newly Infected Excreting Farms	Notes
Week 1	29 March 2017	167	29	138	0	
Week 2	5 April 2017	167	29	138	1	One excreting after Week 1 removed from emitting status given they change outbreak status after Week 1. Another site became infected in Week 2 and started emitting
Week 3	12 April 2017	166 *	35	131	6	Six new sites became infected in Week 3

* Number of participant farms could vary by week. The risk was estimated for the participant farms in the Morrison Swine Health Monitoring Project (MSHMP) of the University of Minnesota
(https://vetmed.umn.edu/centers-programs/swine-program/outreach-leman-mshmp/mshmp, accessed on 20 March 2020) in week 3.

## Data Availability

The data are not publicly available due to privacy concerns. The data presented in this study are available on request from the Morrison Swine Health Monitoring Project (MSHMP; shmp@umn.edu) of the University of Minnesota (restrictions apply).

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
