# Peer review of "Adapting an Atmospheric Dispersion Model to Assess the Risk of Windborne Transmission of Porcine Reproductive and Respiratory Syndrome Virus between Swine Farms"

_viruses, 2022, doi:10.3390/v14081658_

Round 1
Reviewer 1 Report
General comments
Adapting an atmospheric dispersion model to assess the risk of windborne transmission of Porcine Reproductive and Respiratory Syndrome virus between swine farms
The paper uses an atmospheric dispersion model called Hybrid Single-Particle Lagrangian Integrated Trajectory (HYSPLIT) and a web-based platform named Tool for Assessing Pest and Pathogen Aerial Spread (TAPPAS) for modeling windborne PRRSV using PRRSV-specific parameters and a case study based on MSHMP data to demonstrate the value of the model. The model introduced in this paper can assist in outbreak investigations to gauge the likelihood of PRRSV farm-to-farm aerosol transmission.
The paper is very well written and easy to read. Thank you.
I have a few questions about the study, but with revisions and clarifications, it should be published.
Background
The authors do a good job introducing the problem and relevant literature background leading to the objectives. The only issue with the background is a few of the references cited are not listed at the end.
56 – Wei et al., 2016 is not listed in the references
60 – Cox, 1989 is not listed in the references
Data and Methods
158 – 160 – none of the citations used here are listed in the references
163 – Anderson et al., 2017 is not listed in the references
177 – Hess et al., 2008 and Alonso, 2017 are not listed in the references
177 – 179 – The authors mentioned a questionnaire and interview to gather opinions from four experts. What were the questions used and how was that data used in the modeling?
179 – Hsu, 2007 and Kinsley et al., 2016 are not listed in the references
190 – Here the authors introduce the first table to the manuscript; however, that is table 2. Perhaps rework the order of the tables.
194 – what are the implications of not using temperature, humidity, and UV tolerance levels of the virus in the model? Do the authors expect the results to have external validity? I am not familiar with atmospheric dispersion models, but are there other options that would allow incorporating those important parameters? If those parameters are incorporated the model can be much more useful.
195 – 197 – What was the assumption of a two-week window based upon? Reference? Shouldn’t this window be variable and account for variation in parameters such as temperature or other weather conditions? Isn’t the likelihood of an incursion higher over shorter durations (virus viability)?
207 – Archer, 2014 is not listed in the references
208 – Reche et al., 2018 is not listed in the references
210 – Draxler and Hess, 1998 and Draxler and Rolph, 2015 are not listed in the references
225 – 227 – Is the diagnostic criteria for a PRRSV outbreak diagnosis in the MSHMP standard? Do all companies, sites, and veterinarians use the same criteria for reporting an outbreak, or was the outbreak defined exclusively based on previous status and positive PCR testing? Do all sites test at the same frequency or only when clinical signs commonly associated with PRRSV infection are present?
234 – 235 – I suggest the sentence be modified to “The inventory of pigs ranged from 30 to 6,000 with a meat of 1,975 heads”
237 – the number of outbreaks reported for week 2 (04/05/2017) in the manuscript does not match what is presented in Table 1. Which is correct?
236 – “…compared to the number of outbreaks in 03/29/2017…” The word outbreaks does not mean that those were new detections at that moment, but rather that sites were previously diagnosed with a new PRRSV introduction and were considered to be excreting PRRSV, correct? Please, provide the definition of outbreak used in the manuscript and use the same terminology in the tables and manuscript.
Table 1 – in the column “No. non-infected susceptible farms” I get that the total number of farms is displayed after the equal sign, but what is the point of the equation? What do 133 + 6, 132 + 6, and 129 + 6 mean in week 1, 2, and 3, respectively?
269 – 270 – what is the implication of this assumption for the model output? Can the results be extrapolated to farms in other regions, or even applied to the case study farms in the context of temperature and humidity during the week of April 12, 2017?
273 – 275 – this assumption affects the validity of the model significantly.
285 – The authors mention 31 source farms, but table 1 shows 29 farms as infected and excreting. Please, correct this number and make necessary adjustments to the manuscript or correct the table.
286 – how was the number of released particles determined?
296 – Jenks, 1967 is not listed in the references
310 – 312 – This information was previously presented in lines 292 to 298. Please, consolidate those 2 pieces.
312 – 318 – Many of the parameters used in the model come from experimental studies; however, for the infectious dose of aerosol exposure the authors decided to apply examination of histograms to determine values that would be relevant. Why was that choice made?
Results
In the results section, the authors spend a lot of time discussing the results or explaining choices for model parametrization. This is not the place.
337 – what parameters were decided based on an expert opinion? That should be made clear in the material and methods.
353 – 357 – This is one of the major limitations of this work as it is not specific to a season or climatic condition. This point is well made here, but it is misplaced. This should be part of the discussion.
378 – 404 – Although very important for viral viability outside of the pig, none of the parameters discussed in this paragraph are included in the model. I don’t see the need for this paragraph. This information can be used in the discussion to emphasize the importance of not over-interpreting the results given that key parameters were not included in the analysis.
420 – Sorensen et al., 2000 is not listed in the references
440 – 454 – The infectious dose of PRRSV necessary for aerosol transmission was discussed here, but none of this information was taken into consideration for the parameter utilized in the model. I suggest the authors revise this paragraph and justify their choice of 100, 1000, and 10000 mass for emission quantities. This paragraph can be otherwise deleted as some of this information is presented at 314 – 315.
472 – 474 – The authors mention that all six farms received wind from a source farm during the 14-day period. What does “received wind” mean? In figure one all five farms clustered in the southeast of the state do not appear to have any deposition footprint.
Figure 1 – is it possible to highlight source/emitting farms in the figure? Also, please use the same terminology to refer to those farms across the manuscript (source farms, emitting farms, or infected and excreting farms).
485 – what is the difference between n=50 and n=13? From figure 2 it seems that among all possible scenarios (n=18) a total of 50 farms were considered PIFs, but only 13 were unique farms. Is that interpretation correct? If so, I suggest that is incorporated in the manuscripts for bullet points 1, 2, and 3 of this section (484 – 493).
502 – 503 – figure 4 suggests that all PIFs were within 20 km but with a much higher percentage of farms within 10 km from the closest emitting source. It is not possible to observe farms in the 20-25 km distance band. Furthermore, for the 15-20 km distance band only a small percentage of farms was observed to be a PIF based on maximum daily accumulation.
Figure 4 – it seems that in the y-axis the maximum percentage of PIFs can be higher than 1. Is that correct?
Discussion
539 – 540 – those findings are on the conditions of this study, i.e., constant shedding from the infected sow farms over the simulation period and no influence of temperature and humidity on viral survivability. Those are important caveats and should be made clear. Also, please review the 25 km finding.
550 – 551 – perhaps the question that the model answers is “what proportion of farms are at a high risk for PRRSV aerosol transmission?”. As the authors stated previously “…a high concentration of the virus might have been deposited only establishes windborne transmission as a probable pathway for the introduction of pathogen onto a farm.”
555 – check results to see if 25 km is correct and adjust where necessary if necessary.
558 – in the result section it is stated that only one of six farms was within 10 km from a source farm, and therefore not able to rule out windborne PRRSV transmission. Please, correct this section.
560 – cluster of five farms?
579 – 581 – perhaps it should be emphasized that spread over a 10 km distance was uncommon for the conditions explored by the model.
Conclusion
634 – Probably needs to be 20 km under the conditions of this simulation without taking temperature and humidity into consideration.
References
References 10 and 11 are the same
A few references were not used during the manuscript and should be rechecked. Those are 1, 3, 7, 8, 16, 17, 23, 27, 28, 29, 31, 44, 45, 47, 69
55, 56, and 57 may have all been used but are not clearly identified. Please, for authors with more than one publication in the same year identify them by using letters both in the citation and reference.
Author Response
We appreciate the comments and suggestions provided by the reviewer. Our responses to the specific points are included in the word document attached.

Reviewer 2 Report
To predict potential contributions of windborne dispersions of PRRS virus between swine farms an atmospheric dispersion model was employed. Overall, the work was described in detail, and a significant survey of literature parameters was performed to inform the modeling. A case study based on a 2017 two-week outbreak highlighted the need to consider the possibility for greater dispersion (25km radius) than commonly assumed (10 km). More specific points to consider follow: 1) Input from experts Although this input was mentioned and cited, the actual ways such input influenced the modeling was not described. Please include some description of the role the expert inputs affected how the model was formulated or implemented. 2) Reproducibility of results How much work will others (outside of the co-authors of the current project) need to invest to reproduce the findings or at least carry out the case study on their own? A growing area of concern and opportunity is the need to develop quantitative mechanistic models that can be readily reproduced, ideally by sharing code (e.g., Github, project link, or lab website) and seeking ways to ensure the long-term availability of the data, models and code. My preliminary assessment is that the work would not be easily reproduced, limiting its impact and ability to serve as a foundation for adoption or extension by others. Revision should address this issue. Ideally, interested readers could easily download and run the model to reproduce findings or easily alter individual model parameters to explore their effects. For details, see Erdemir, Ahmet, et al. "Credible practice of modeling and simulation in healthcare: ten rules from a multidisciplinary perspective." Journal of translational medicine 18.1 (2020): 1-18. Line 443 Fix exponents on TCID_50 measures Line 566-567 Fix unclear sentence "This indicate focusing on the importance of focusing…"
Author Response
We appreciate the comments and suggestions provided by the reviewer. Our detailed response is attached.

Reviewer 3 Report
Brief Summary and Strengths
In the following manuscript the authors focus on PRRSv, a virus that causes large revenue losses in the swine agricultural industry. While all members of the field are not in agreement on the transmission limitations of this virus, it is noted that previous evidence supports the potential for the virus to be spread by windborne dispersal and that during outbreaks it would be useful to understand how relevant this method of transmission should play into mitigation strategies. In order to address this issue, the authors aimed to create an atmospheric dispersion model using reported atmospheric data and pairing that with data on known infections that occurred during a PRRSv outbreak in Minnesota, USA in 2017. While there are current limitations of reporting PRRSv infections within the industry, this model would provide parameters that could improve PRRSv oubreak management. The reported model soundly encapsulates many factors, including predictors of particle movement and virus stability. This initial study concluded that the created model would recommend monitoring all farms within 25 km of those reporting PRRSv infections rather than the previous guidelines of 10 km, a more conservative approach aimed to reduce infections caused by windborne dispersal of PRRSv.
General Comments
The major weakness of the manuscript is the use of only one case study, however, as stated by the authors, there is no mandate to report the data and procurement was only available through MSHMP, a voluntary reporter program. While the manuscript is limited to predicting scenarios where windborne dispersal of virus could occur, the authors note several times in the discussion of what their model does not tell them, most importantly, confirmation of airborne PRRSv infection rate.
Specific comments
Line 561 In the sentence including “a common route of pathogen introduction that is not capture by the current model,” the authors need to provide a robust list of factors their model does not account for and how they could limit the model.
Author Response

(The authors gave the same response as above.)

Round 2
Reviewer 1 Report
This reviewer thanks the authors for their corrections and effort in improving the paper. The paper was improved substantially from a high starting quality.
This reviewer has no further comments about the following sections: Title, abstract, background, data, and methods.
I only have a few points about the results and discussion. A few of those comments need to be addressed before a recommendation for publication can be made.
Results
547 – 551 – I appreciate the change in Figure 1 showing the deposition footprint after 24h and 36h. Could those be expanded to figures in full scale? I still have two questions regarding the results of this simulation. 1) It is now clear that there is deposition in all six farms having new outbreaks during the study period; however, according to the parameters in the model, did all farms receive a critical mass of virus enough to cause an outbreak or at least for them to be considered PIFs? And 2) in Figure 4 authors comment that all PIFs were within a 25-km of the closest emitting farm, and in this sentence, you state that the farms in the southeast cluster were >57 Km away from any of the emitting farms. How do you reconcile those two findings?
594 – 596 – “All these uniquely identified PIFs were within 25 km of the closest emitting farm (Figure 4), with 53.66% within 10 km.” I understand there were 13, 41, and 105 unique PIFs for the deposition thresholds using cumulative 14-day deposition, 14-day median daily deposition, and maximum daily deposition, respectively. Figure 4 shows many PIFs for each category that would be higher than the abovementioned counts. What numbers are the authors using to calculate the percentage of PIFs in each distance band? Please, provide a table in the supplementary material that specifies those numbers as it is unclear.
Discussion
659 – I suggest the authors let the use of the information generated by this work dictate how much progress and the contribution it will give to the swine industry.
667 – I would still like to see the breakdown of information regarding the distance of PIFs before 25 km is proposed as the assigned distance for potential PRRSV airborne transmission. Furthermore, this observation has to be prefaced or mention the following caveats: the work done is model-based using a set of parameters, of which not all are validated. Moreover, the model lacks the input of essential weather information such as temperature and humidity. Therefore, 25 km PRRSV airborne transmission may be possible under the conditions of this study.
Once the points mentioned above have been clarified, changes may be necessary throughout the manuscript.
Author Response
We appreciate the suggestions the reviewer has provided and appreciate their time. Attached are our responses to the comments. Necessary edits have been done to the manuscript including language, spell checks, and figures.

Reviewer 2 Report
The authors have adequately responded to my previous concerns.
Author Response
We appreciate the suggestions the reviewer has provided and appreciate their time. The manuscript has been revised for language, style, and spell check.